# Density Distribution in Wood of European Birch (*Betula pendula* Roth.)

**Ewa Dobrowolska, Paulina Wroniszewska and Agnieszka Jankowska *** 

Institute of Wood Sciences and Furniture, Warsaw University of Life Sciences – SGGW, Nowoursynowska 159, 02-787 Warsaw, Poland; ewa_dobrowolska@sggw.pl (E.D.); paulina_wroniszewska@sggw.pl (P.W.)

*   Correspondence: agnieszka_jankowska@sggw.pl; Tel.: +48-22-5938634

**Abstract:** The aim of the presented research is to perform a comprehensive analysis of wood density variability on the longitudinal and transverse cross-section of log trees at the age of 70 to 72 years of *Betula pendula* Roth. and the creation of density distribution maps. Furthermore, the determination proportion of juvenile and mature wood was done. Wood density was determined with a non-destructive method using an isotopic densimeter. It was found that the wood location, both in cross-section and longitudinal section of the trunk, had a statistically significant effect on the average density of birch wood. The average density of whole logs was significantly higher than the average density at the breast height. On the cross-section, the distribution of average densities determined at the breast height, as well as on 1/4 of the log height, properly depicted the distribution of average densities on the cross-section determined for the whole logs. The geographical direction (north–south) did not have a statistically significant effect on the distribution of average densities on the cross-section of the tested birch logs.

**Keywords:** density distribution; European birch; *Betula pendula* Roth.

## 1. Introduction

Most of the basic physical, mechanical, and technological properties of wood depend on its density. Synthetically, it expresses many features related to the wood anatomy as well as physical and chemical structures [1]. The density is influenced by a number of factors such as wood species, wood moisture content, annual growth rings and proportion of late-wood, age of the tree and its height, location within the trunk's longitudinal and longitudinal cross-section, as well as habitat conditions [2–7]. From the analysis of numerous studies on the wood density of various species, the relationship between the width of annual growth and density was not found in the case of deciduous wood species [2,3,5,7–9]. Additionally, the independence from the cross-sectional area of the trunk was confirmed [5]. It is also believed that the wood density of these species increases from trunk pith to the bark [3]. According to Krzysik [2] and Niemz [9], the density minimum of deciduous trees is associated with juvenile wood. The influence of the proportion of juvenile wood, mature wood, and wood overridden on the distribution of wood tissue density on the cross-section was also found. Juvenile wood is a part of trunk created by cambium cells in the first years of the life of the tree and constitutes the central part in the form of a cylinder, consisting of several or even several dozen of annual growth rings [10]. The width of this zone is influenced by many factors such as species, genetic conditions, tree growth rate, type of habitat, planting, distance and age of the cambium, geographical location, breeding conditions, as well as the place of the tree in the stand [6,10–14]. According to Hakkilla [11], the proportion of juvenile wood should be considered during the analysis of wood density. The identification of the juvenile wood zone mainly consists of the characterization of structural elements of wood tissue.

In the ring-porous wood species, vessels in juvenile wood are distributed over the entire width of the annual rings as in case diffuse-porous wood species. Another feature that occurs in juvenile wood is the type of perforation in the vessels. In juvenile wood vessels with a simple perforation, ladder perforation was noticed [10]. Mature wood is created by cambium cells when they reach their maximum dimensions. This part of the wooden tissue is characterized by relatively constant cell dimensions and properties [15]. Compared with mature wood, juvenile wood usually stands out by the smaller length of anatomical elements and the proportion of late-wood, which is associated with lower density and worse mechanical properties [6,14,16–18]. Depending on the class and wood species, the strength of juvenile wood constitutes from 50% to 70% of the strength and hardness of mature wood [19].

Wood density is one of the most important technological parameters defining its industrial use. The popular species list includes birch wood (*Betula*) from the birch family (*Betulaceae*) [20], occurring in almost all of Europe, with the exception of Spain, Greece, and the southern part of Italy, as well as in parts of Asia Minor, Caucasus, and western Siberia. Birch is a pioneer species, rapidly growing and plays an important role in the early stages of primary and secondary succession, as well as in preparing areas for settlement by more demanding species [21]. Moreover, birch belongs to the basic species in plantations of fast-growing forest trees [22]. That wood species is used in industrial applications for the production of plywood, veneers, and furniture, and is a valuable raw material for the pulp, paper, and board industry [3,23]. Due to the significant economic importance of birch wood, many studies on the distribution of density in birch wood have been found in the literature, particularly depending on its age and diameter [7,24,25]. However, so far, extensive research regarding birch wood density has been conducted on relatively young wood dominated by juvenile wood zone. Only a study on wood from mature Finnish trees has been found [26].

The aim of this research is to perform a comprehensive analysis of wood density variability on the longitudinal and transverse cross-section of log trees at the age of about 70 years of European birch (*Betula pendula* Roth.) and the creation of density distribution maps. Moreover, the determination proportion of juvenile and mature wood was made, as well as determination of the size and range of knots in both zones. Wood density was determined with a non-destructive method using an isotopic densimeter.

## 2. Materials and Methods

For research, five birch (*Betula pendula* Roth.) logs were used, originating from north-eastern Poland. The trees were 70 to 72 years old, from the first bonitation and from the first biosocial class, according to Kraft, the so-called towering trees. The height of the trees was about 22 m, with a crown index of 0.3 and diameter at breast high approximately 42 cm (Table 1).

Trees were divided into the logs, which have been sawn with respect to the geographical directions (north, south). Directly to the research, core timber were selected in the total number of 69. After drying and planing, the dimensions of the samples were width approximately 70 mm, length approximately 1.5 m, and the moisture content was 12%.

**Table 1.** Characteristics of experimental trees of silver birch (*Betula pendula* Roth.).

| Feature | Log Number | | | | |
|---|---|---|---|---|---|
| | I | II | III | IV | V |
| Age * (years) | 71 | 72 | 72 | 70 | 70 |
| Diameter at breast high in bark (cm) | 42 | 42 | 42 | 42 | 42 |
| High (m) | 22.8 | 21.5 | 22.0 | 21.5 | 21.5 |
| Crown index ** | 0.3 | 0.3 | 0.3 | 0.3 | 0.3 |

* Based on the number of growth rings on the cross-section of the lowest part; ** the ratio of the length of the crown to the height of the tree.

The density of wood was determined using the MGD-05 isotope density meter (resolution 0.1 kg/m$^3$) by using energy-transmission gamma 59.5 keV (Amerycium $^{241}$Am). The diameter of used radiation collimator was 5 mm. The distance of the radiation source from the detector was 150 mm. The time of a single measurement was 30 s. The result of a single measurement was subject to an error ±10 kg/m$^3$ [27]. The diagram of the isotope X-ray density meter MGD-05 is given in Figure 1.

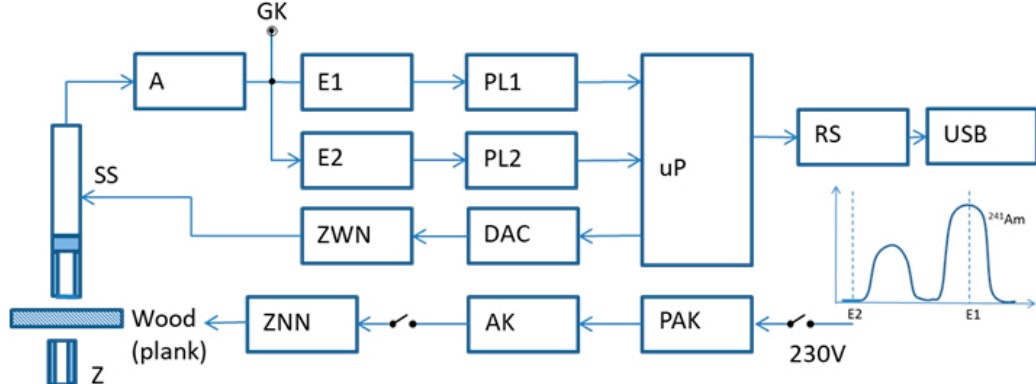

**Figure 1.** Diagram of the isotope X-ray density meter MGD-05, where Z—source of radiation (241Am), SS—scintillation probe, A—amplifier of pulses, E1–discrimination threshold of automatic gain control, E2—measurement path discrimination threshold, PL1—pulse counter of the measuring path, PL2—impulse counter of automatic gain adjustment, uP—microprocessor, DAC—digital to analog converter, ZWN—high voltage power supply, ZNN—low voltage power supply, RS—serial port, USB—USB interface, AK—accumulator, PAK—charging rectifier.

Examination of the density distribution of the cross-section of logs was carried out in two directions (in the north and south geographical directions) along a measuring lines (parallel to the longitudinal axis of the timber) spaced in distance 20 mm. The density distribution on the longitudinal section was examined along measuring lines spaced in distance 150 mm, taking into account defects such as knots, cracks, false heartwood, stoppers. The density maps were prepared using the MATLAB computer program.

Determination of the proportion of juvenile and mature wood was performed by the measurements of annual growth rings width and wood fibers length. The measurements were made on samples taken along the northern radius, using samples obtained from the breast height.

The width of the annual increments was marked using the BIOTRONIC electronic gravimetric increment. Measurements were made with an accuracy of 0.01 mm on strips whose surfaces were previously leveled with a microtome knife. From annual increments of 3, 6, 9, 12, 15, 20, and beyond, from every fifth to the perimeter, macerated wood samples were prepared to assess the length of wood fibers. The measurements were performed using a MICROSCAN Imager 512 computer image analyzer with an accuracy of 0.1 μm.

The obtained results (average values from the obtained measurements of the properties of the examined logs) were subjected to statistical analyzes including statistical measures [28,29], one-factor and two-factor analysis of variance and Tukey's test, as well as two-tailed Student's *t*-test and two-segment regression. The STATISTICA PL computer program and the Microsoft Office Excel 2010 program were used directly for the statistical processing of results.

## 3. Results and Discussion

### 3.1. Determining the Proportion of Juvenile and Mature Wood

The characteristics of the annual growth width were made at the breast height. The results are presented in Table 2. The average annual increase width was 2.6 mm with the coefficient of variation

equal to 47%. Table 3 presents the results of the length of fiber measurements. The average length of fibers was 1279 µm (coefficient of variation was 20%). The average length of fibers was within the range of numerical values contained in the literature [29–31].

**Table 2.** Characteristics of annual growth (at the breast height) of birch debts.

| Feature | Value |
|---|---|
| Average width | 2.6 mm |
| Coefficient of variation | 47% |
| Minimal width | 0.4 mm |
| Maximal width | 6.5 mm |
| Average number of annual rings | 68 |

**Table 3.** Characteristics of fiber length.

| Feature | Value |
|---|---|
| Average length | 1279 µm |
| Coefficient of variation | 20% |
| Minimal length | 440 µm |
| Maximal length | 2032 µm |
| Number of measurements | 1380 |

Determining the proportion of juvenile and mature wood was carried out based on the measurement of the length of wood fibers. The obtained results were analyzed using two-segment regression [32,33], in which the adjustment of the function in two segments is optimal and the overall estimation error is as small as possible. On this basis, a juvenile wood zone was detected, comprising the first 25 annual growth rings. The remaining part was mature wood (Figure 2). According to Bonham and Barnett [34], the juvenile wood zone in *Betula pendula* Roth. wood covers the first 15 annual growth rings. Differences between the results justify the taking surveys on wood from different types of habitats or different regions.

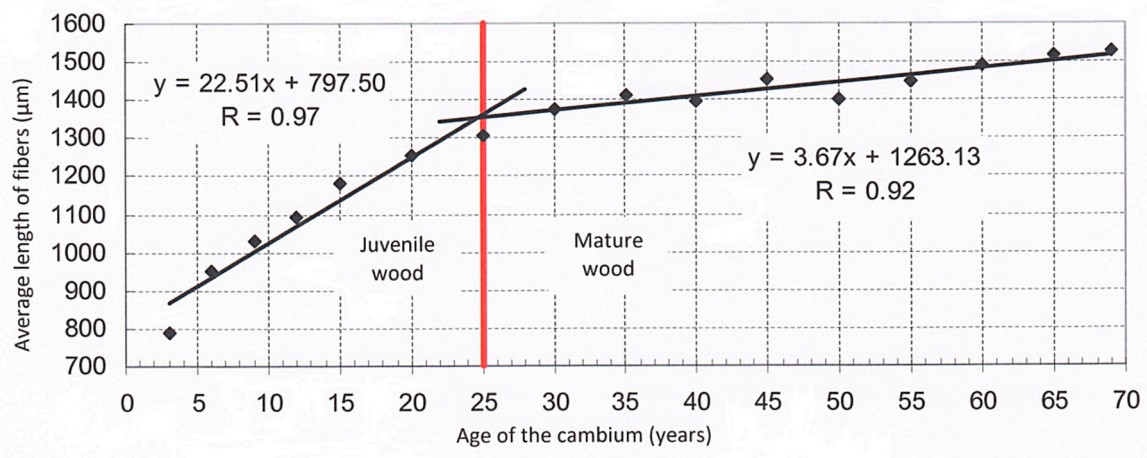

**Figure 2.** Juvenile and mature wood in the examined birch wood.

The obtained results were not in accordance with numerical values known from the Polish literature [35], where the size of the juvenile wood zone in birch trees was estimated as 14 annual increments, forming a cylinder with a diameter of 12 to 13 cm, covering the area from 30% to 60% of the cross-section. In the tested birch trees, juvenile wood at the breast height was in the formed a cylinder with a diameter of about 16 cm. The proportion of the juvenile wood zone covers approximately 15% of the cross-section.

In the zone of juvenile wood, fiber length increased from about 800 (in core) to 1300 mm (an increase of over 60%), and, in the mature wood zone, the increase was about 17%.

### 3.2. Wood Density Distribution Analysis

Results of average density at 12% moisture content are given in Table 4. The average density of tested silver birch wood was 623 kg/m$^3$.

**Table 4.** Results of tests on average density of tested silver birch (*Betula pendula* Roth.) at 12% moisture content.

| Density | |
|---|---|
| Average density (kg/m$^3$) | 623 |
| Standard deviation (kg/m$^3$) | 58 |
| Variability coefficient (%) | 9.3 |
| Average density at breast high (kg/m$^3$) | 608 |
| Average density at 1/4 high of the log (kg/m$^3$) | 630 |
| Minimal value of density (kg/m$^3$) | 318 |
| Maximal value of density (kg/m$^3$) | 895 |
| Difference between maximal and minimal value of density (kg/m$^3$) | 577 |
| Number of measurements | 8320 |

According to Hakkila [11,36], conventional density in case of pine, spruce, and birch is close to density at 1/4 high of the log. But according to other research [37–39], wood density and other parameters determined from wood at breast height represent average values for all logs. Due to discrepancy of information in this area, a two-sided *t*-test was done (Table 5). The results of statistical analysis confirmed that there was no difference between average density of all logs and average density at 1/4 high of the log.

**Table 5.** Two-sided *t*-test of the difference between average densities (at significance level $p = 0.05$) for density distribution in birch wood.

| Parameter | Empirical Statistics $t$ | Critical Statistics $t_{0.05/2; v}$ | Number of Degrees of Freedom $v$ |
|---|---|---|---|
| Average density of all logs/Average density at breast height | **2.413** | 1.960 | 8401 |
| Average density of all logs/Average density at 1/4 high of the log | 0.860 | 1.993 | 73 |

### 3.3. Density Distribution of Silver Birch Wood on the Cross Section

The density distribution of all logs of silver birch wood on the cross-section is given in Figure 3. The results of a two-factor analysis of variance (Table 6) showed the following relations:

- geographical direction (north–south) did not have a statistically significant effect on the distribution of average densities across the entire birch log;
- distance from the pith had a statistically significant effect on the distribution of average densities across the entire birch log;
- there was no significant interaction of geographical direction and distance from the pith.

Miler [40] also showed in his research the insignificance of the influence of the geographical direction on the density of birch wood.

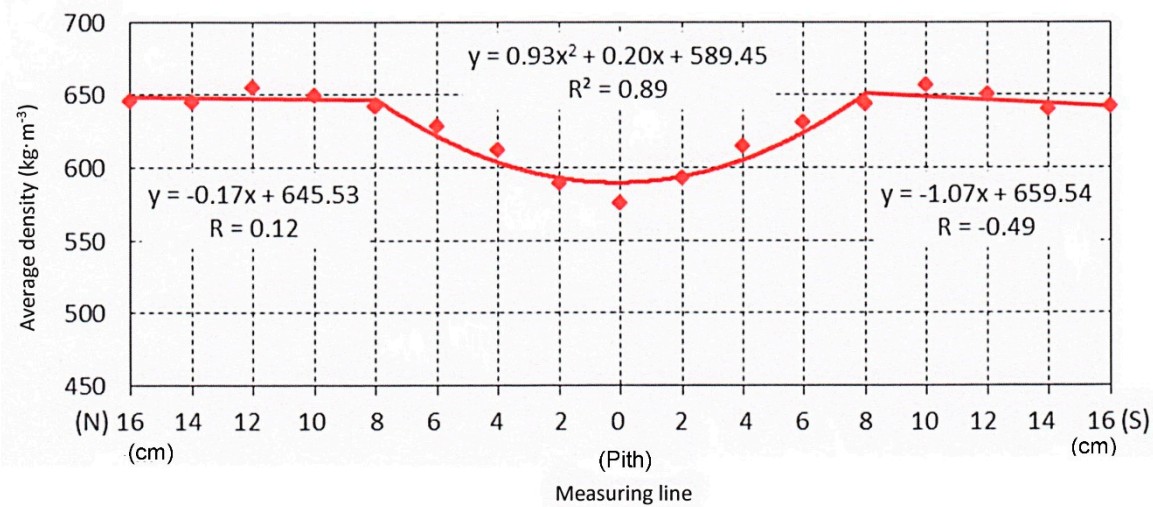

**Figure 3.** Density distribution of all logs of silver birch wood on the cross-section.

**Table 6.** Two-factor analysis of variance (at the significance level $p = 0.05$) for the distribution of average densities across the entire birch log.

| Source of Variation | Sum of Squared Deviations | Number of Degrees of Freedom | The Mean Square of Deviations | Empirical Statistics F | Test Probability $p$ |
|---|---|---|---|---|---|
| Intercept | 2,039,755,000 | 1 | 2,039,755,000 | 784,716 | 0.000 |
| Geographical direction | 299 | 1 | 299 | 0.115 | 0.735 |
| Distance from the core | 3,709,976 | 7 | 529,997 | 203.896 | 0.000 |
| Direction·Distance | 22,208 | 7 | 3173 | 1.221 | 0.287 |
| Error | 19,786,270 | 7612 | 2599 | - | - |

Multiple comparisons were made using medium Tukey's test in order to verify the levels between which the factor "a distance from the pith" had a significant difference (Table 7). Five homogeneous groups were obtained. The highest average density of 653 kg/m$^3$ was obtained for measuring lines 10 and 12 cm away from the pith, while the lowest average density of 591 kg/m$^3$ was obtained for the measuring line 2 cm away from the pith. The relationship between wood density and tree was also confirmed during studies on silver birch growing at the lower altitude of the Czech Republic region [7] and the Lativian plantation on post-agricultural lands [24].

**Table 7.** Tukey test of the factor "distance from the core" for the distribution of average densities over the cross-section of the entire birch log.

| Distance from the Core (cm) | Average Density (kg/m$^3$) | Test Probability $p$ | | | | | | | |
|---|---|---|---|---|---|---|---|---|---|
| | | 2 | 4 | 6 | 8 | 10 | 12 | 14 | 16 |
| 2 | 591 | | 0.000 | 0.000 | 0.000 | 0.000 | 0.000 | 0.000 | 0.000 |
| 4 | 613 | 0.000 | | 0.000 | 0.000 | 0.000 | 0.000 | 0.000 | 0.000 |
| 6 | 630 | 0.000 | 0.000 | | 0.000 | 0.000 | 0.000 | 0.000 | 0.004 |
| 8 | 643 | 0.000 | 0.000 | 0.000 | | 0.000 | 0.004 | 1.000 | 1.000 |
| 10 | 653 | 0.000 | 0.000 | 0.000 | 0.000 | | 1.000 | 0.007 | 0.289 |
| 12 | 653 | 0.000 | 0.000 | 0.000 | 0.004 | 1.000 | | 0.026 | 0.411 |
| 14 | 643 | 0.000 | 0.000 | 0.000 | 1.000 | 0.007 | 0.026 | | 1.000 |
| 16 | 644 | 0.000 | 0.000 | 0.004 | 1.000 | 0.289 | 0.411 | 1.000 | |
| Homogeneous group number | | 1 | 2 | 3 | 4 | 5 | 5 | 4 | 4. 5 |

For comparison, the distribution of average densities across the cross-section at breast height and 1/4 of the height of the birch log was examined (Figures 4 and 5). At breast height and at 1/4 of the

height of the birch logs in cross-section, the average density of wood increased along the radius of the tree from the core to a distance of about 8 cm from the core, and then there was very little variation in density.

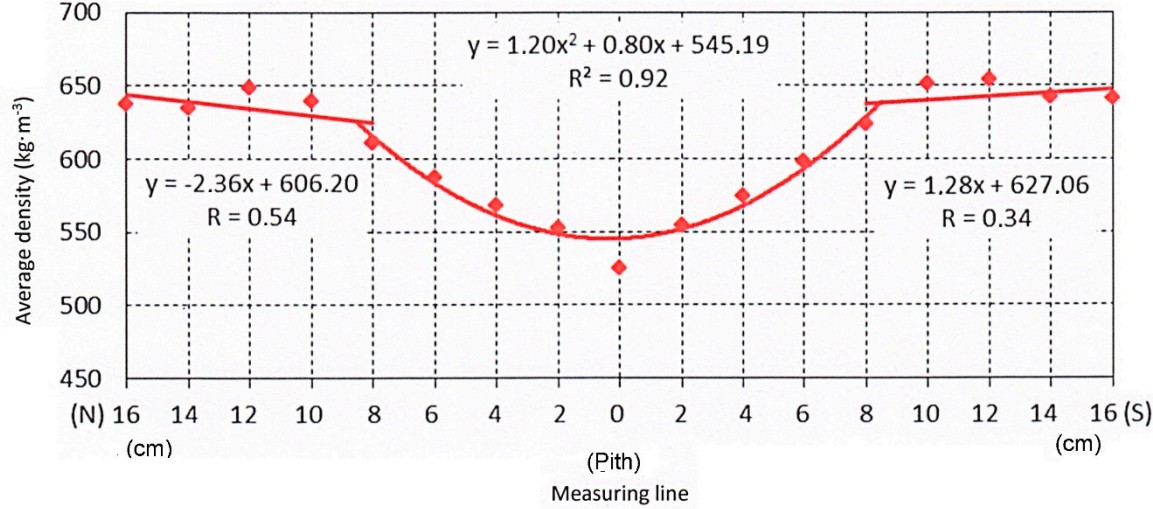

**Figure 4.** Distribution of average densities on the cross-section of birch log at breast height.

Two-sided tests of significance of correlation coefficients showed that, in both cases, the correlation was statistically significant (Table 8). The distribution of average densities across the cross-section determined both at breast height and 1/4 of the log height properly depicted the distribution of average densities across the cross-section determined for the entire birch log.

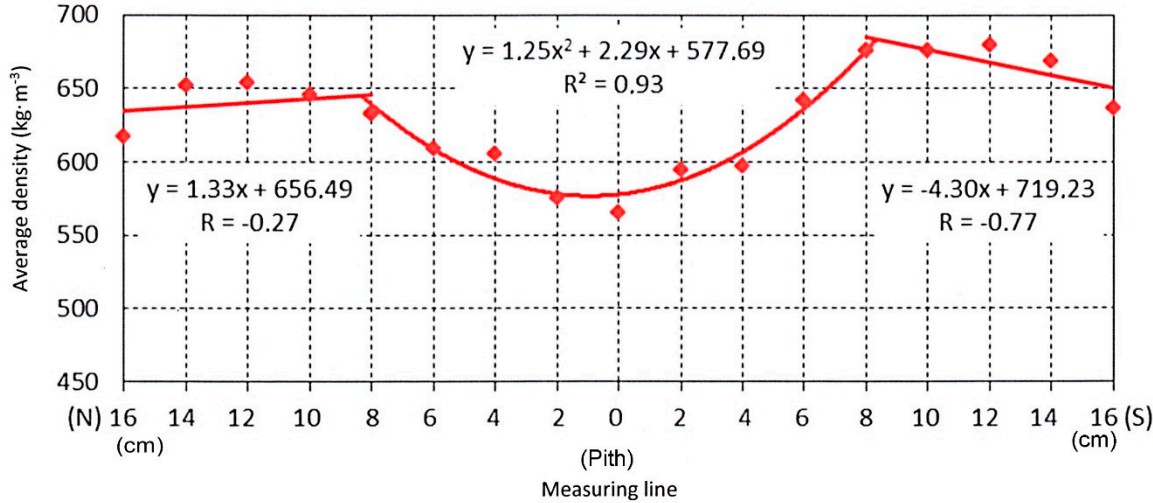

**Figure 5.** Distribution of average densities on the cross-section of a birch log at 1/4 of the log height.

**Table 8.** Two-sided tests of the significance of correlation coefficients between average densities across the entire birch log and average densities across the cross-section at breast height and 1/4 of the height of the birch log (at the significance level $p = 0.05$).

| Area Type | Correlation Coefficient $R$ | Critical Values $R_{0.05/2; v; k+1}$ | Number of Degrees of Freedom $v$ |
|---|---|---|---|
| Distribution at breast height | **0.977** | 0.666 | 7 |
| Distribution at 1/4 of the height of the log | **0.959** | 0.666 | 7 |

The radial density increase (average for two geographical directions) in the juvenile wood zone for the entire birch log was 11.9% (Table 9).

**Table 9.** Radial density increase (in the juvenile wood zone) on the cross-section of the birch log.

| Area Type | Radial Density Increase (%) |
|---|---|
| All logs | 11.9 |
| At breast height | 17.7 |
| At 1/4 of the height of the log | 15.9 |

### 3.4. Density Distribution on the Longitudinal Section of a Birch Log

Figures 6 and 7 show the distribution of average densities on the longitudinal section of a birch log.

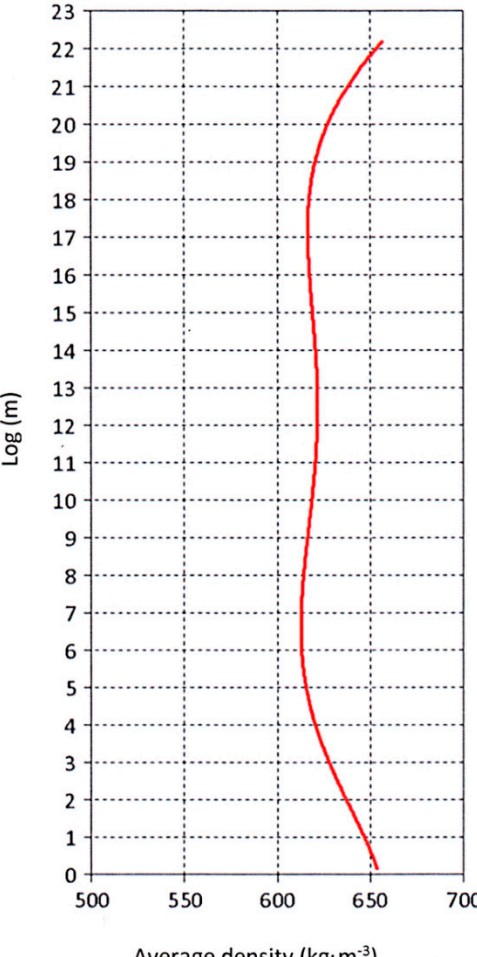

**Figure 6.** Distribution of average densities on the longitudinal section of a birch log.

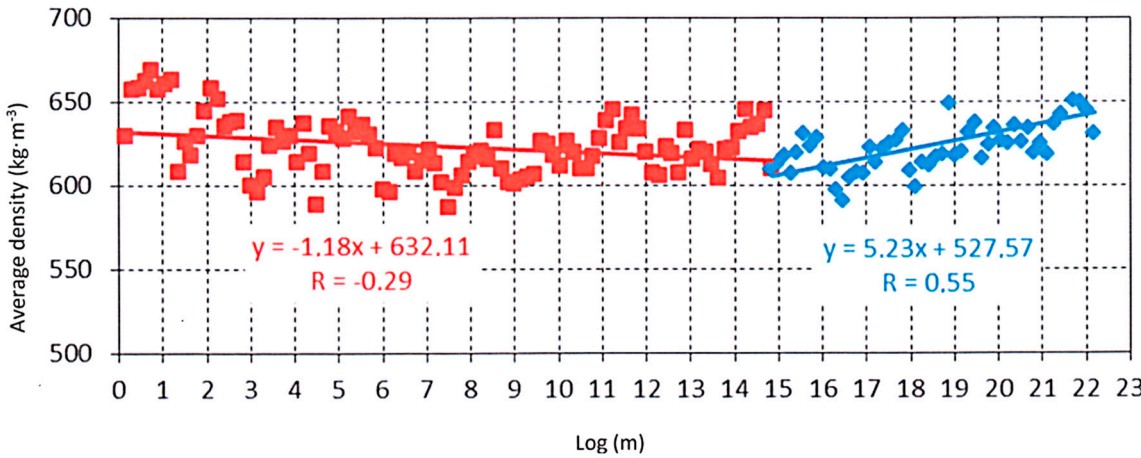

**Figure 7.** Distribution of average densities on the longitudinal section of a birch log.

Based on the results of one-way analysis of variance (Table 10), it was found that the location of the wood at the trunk height had a statistically significant effect on the average density of birch wood. That relation was also observed in silver birch logs from Lativia [24], even though the analyzed wood consisted mainly of juvenile wood what is opposite to material used for presented research.

**Table 10.** One-way analysis of variance (at significance level $p = 0.05$) for distribution of average densities on the longitudinal section of birch logs.

| Empirical Statistics $F$ | Critical Statistics $F_{0.05;\ u;\ v}$ | Number of Degrees of Freedom $u,\ v$ |
|---|---|---|
| **5.602** | 1.202 | 147. 8172 |

Correlation coefficients were determined and two-sided tests of significance of correlation coefficients were verified, checking the significance of the statistical correlation between the average density of birch wood and its location at the height of the trunk. The calculations were carried out for two areas: from the butt to the base of the crown and for the area between the base of the crown and the top of the tree (Table 11). On the longitudinal section of the birch log, the average density decreased as it moved from the tree butt to the root of the crown (a statistically significant correlation was obtained), while in the crown area the average density of wood increased as it moved to the top of the tree (a statistically significant correlation was obtained).

**Table 11.** Two-sided tests of the significance of correlation coefficients between the average density of birch wood and its location at the height (at the significance level $p = 0.05$).

| Area Type | Correlation Coefficient $R$ | Critical Values $R_{0.05/2;\ v;\ k+1}$ | Number of Degrees of Freedom $v$ |
|---|---|---|---|
| From the butt to the base of the crown | **−0.291** | 0.198 | 97 |
| Between the base of the crown and the top of the tree | **0.551** | 0.279 | 48 |

The trunk height was determined (based on the analysis of the accuracy of function matching, by maximizing the value of the correlation coefficient), to which the average density of the tested birch wood decreased as it moved from the butt to the crown root. It amounted to 4.65 m, which was 0.2 times the height of the log. Correlation coefficients were calculated for two areas: below and above 0.2 log height (Table 12). Two-sided tests of significance of correlation coefficients showed that in both areas, the correlation was statistically significant.

**Table 12.** Two-sided tests of significance of correlation coefficients between the average density of birch wood and its location at trunk height below and above 0.2 log height (at significance level $p = 0.05$).

| Area Type | Correlation Coefficient $R$ | Critical Values $R_{0.05/2;\ v;\ k+1}$ | Number of Degrees of Freedom $v$ |
|---|---|---|---|
| Below 0.2 log height | **−0.649** | 0.355 | 29 |
| Above 0.2 log height | **0.306** | 0.182 | 116 |

Figure 8 shows the distribution of average densities on the longitudinal section of a birch log below and above 0.2 log height.

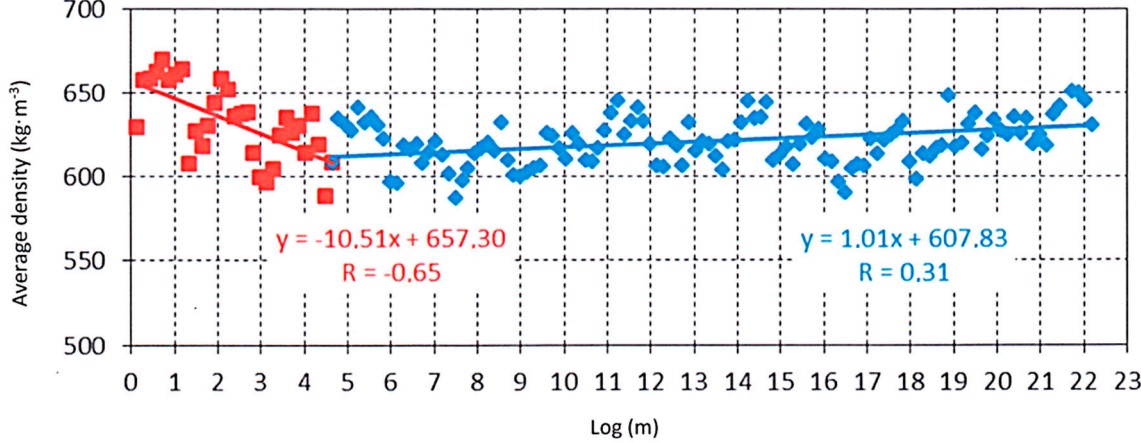

**Figure 8.** Distribution of average densities on the longitudinal section of a birch log below and above 0.2 log height.

The highest density on the longitudinal section was characterized by the butt part of the birch log, confirming the literature data. The average densities on the longitudinal section of the birch log varied in the range 590–670 kg/m$^3$.

### 3.5. Density Map

The literature related to the research on density distribution in experimental tree trunks made by destructive methods (mainly stereometric) characterizes the technical possibilities of rational industrial use of wood. The application of the isotope densimeter to measure the density using the X-ray (non-destructive) method allowed for very accurate determination of the density distribution on the longitudinal and cross-section of the birch logs tested. The density distribution (at 12% moisture content) averaged for all tested logs in the form of a map is presented in Figure 9.

On the map illustrating the density distribution, wood with the highest density not caused by knots (700–750 kg/m$^3$) appeared in the butt part (dark yellow). The correlation between the wood density and the maturity of cells was observed. It was noticed that between the measuring lines spaced 4 cm from the pitch, there was a strip of wood with a density of 550 to 599.9 kg/m$^3$ (light blue). The density of wood behind this area was from 600 to 649.9 kg/m$^3$ (green). This part around the pitch with a radius of about 8 cm formed a zone of juvenile wood with a lower density visible in the form of blue and green colors. Behind the measuring lines spaced 8 cm from the pith (mature wood), there was wood with a density in the range of 650 to 700 kg/m$^3$ (light yellow stripes). Around 11 m in height, near the peripheral parts, a reduced density area appeared (dark blue and navy blue). This phenomenon could be explained by the occurrence of wedges and cracks. At two heights: 18.0 and 21.6 m, wood density was found to be 750 to 899.9 kg/m$^3$ (orange and red). The increase in density in these areas was caused by a significant number of high-density knots. The occurrence of defects, mainly in the

form of knots, had a statistically significant effect on the average density of the large-dimensioned birch wood tested. The average density of wood with defects was higher by approximately 5 kg/m$^3$ compared to the density of wood without defects. The slight difference is due to the significant share of large-sized wood in the knotless zone.

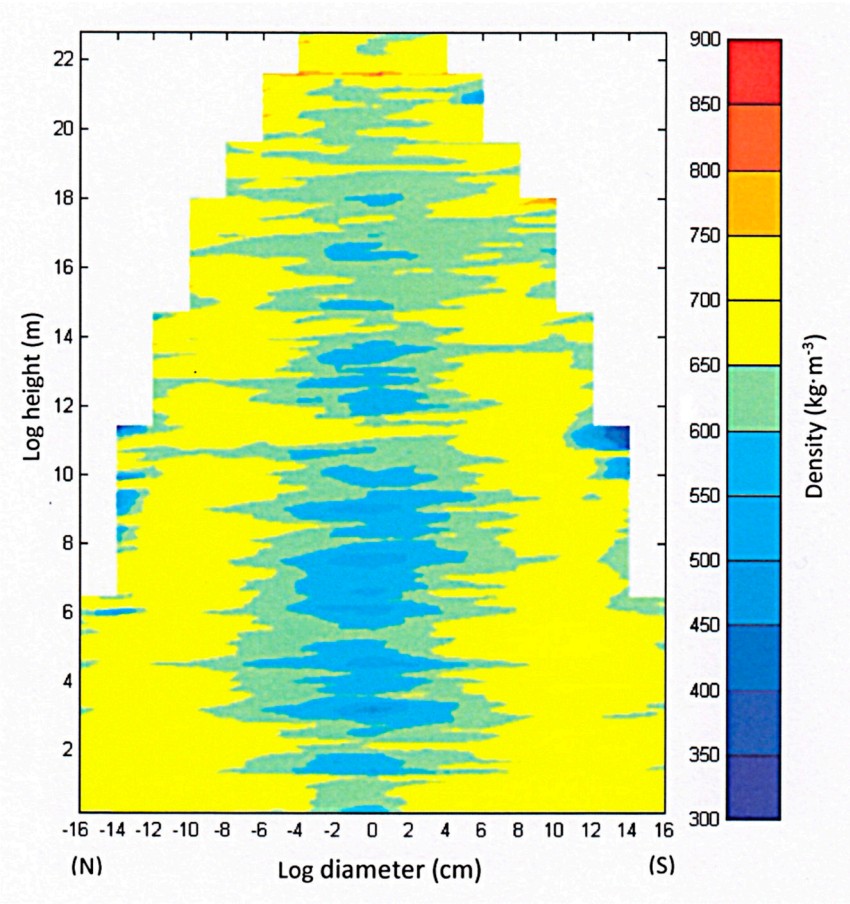

**Figure 9.** Map of density distribution in the studied wood of silver birch.

In birch wood with an average density of 623 kg/m$^3$ (Table 13), over 22% of average density measurement results ranged from 500 to 599.9 kg/m$^3$, and over 75% of average density measurement results ranged from 600 to 699.9 kg/m$^3$.

**Table 13.** Proportion of density measurement results in density ranges.

| Feature | | |
|---|---|---|
| Average density (kg/m$^3$) | | 623 |
| | Density ranges (kg/m$^3$) | Proportion (%) |
| | 0–99.9 | 0.0 |
| | 100–199.9 | 0.0 |
| | 200–299.9 | 0.0 |
| The proportion of average results of density | 300–399.9 | 0.3 |
| measurements in density ranges | 400–499.9 | 0.7 |
| | 500–599.9 | 22.3 |
| | 600–699.9 | 75.4 |
| | 700–799.9 | 1.3 |

## 4. Conclusions

Based on the analysis of density variability of the tested birch wood (*Betula pendula* Roth.) at the age of 70 to 72 years, the following conclusions were made:

1.  The place where wood was tested, both in cross-section and longitudinal section of the trunk, had a statistically significant effect on the average density of birch wood.

2.  The average density of whole logs was statistically significantly higher than the average density at breast height, while the average density at 1/4 height did not differ statistically significantly from the average density of the whole log.

3.  On the cross-section, the distribution of average densities determined at breast height, as well as on 1/4 of the log height, properly depicted the distribution of average densities on the cross-section determined for the whole logs. The distribution of average densities in the cross-section can be described by a second degree polynomial in the juvenile wood area and two straight lines (in the mature wood areas); in the juvenile wood area, the average wood density increased significantly along the radius of the tree; in the mature wood areas, there was a slight density fluctuation.

4.  On the longitudinal section of the tested birch wood, the average density decreased going from the tree butt to 20% of the log height. In the crown area, the average wood density increased as it moved to the top of the tree. In both areas, a statistically significant correlation was obtained between the average density of wood and its location at the height of the trunk.

5.  The geographical direction (north–south) along which the density was determined did not have a statistically significant effect on the distribution of average densities on the cross-section of the tested birch logs.

**Author Contributions:** Conceptualization, E.D. and P.W.; methodology, E.D. and P.W.; investigation, P.W.; resources, E.D. and P.W.; data curation, E.D. and P.W.; writing—original draft preparation, E.D.; writing—review and editing, A.J.; visualization, P.W. and E.D.; supervision, E.D and A.J. All authors have read and agreed to the published version of the manuscript.

**Funding:** This research received no external funding.

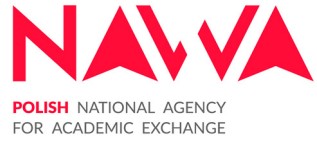

**Acknowledgments:** Publication of the article has been financed by the Polish National Agency for Academic Exchange under the Foreign Promotion Programme.

**Conflicts of Interest:** The authors declare no conflict of interest.

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
