# Peer review of "Density Distribution in Wood of European Birch (Betula pendula Roth.)"

_forests, doi:10.3390/f11040445_

Round 1

Reviewer 1 Report

This article investigates the variation in density of birch wood with position in the log. The results are presented clearly for the most part, are supported by statistical analysis, and should provide useful information for utilization of birch wood. The article is recommended for publication after some improvements in clarity.

Line 38:  Please clarify the meaning of “wood overridden…”

Lines 71-72: This mentions determination of the size and range of knots; however, information about knot size is missing from the results section.

Line 86:  “planing” rather than “planning”

Lines 96-97:  Please clarify how defects were taken into account. Were defects included or excluded?

Lines 141-142:  The juvenile wood zone is given as a percentage of the log diameter rather than a percentage of cross-sectional area.  This is confusing because line 139 mentions literature values and quotes the area of the cross-section.

Figures 2,3,4:  The x-axis is lacking units (cm).

Figure 5:  Is the curve in this figure based on a mathematical smoothing function?  If so, please provide details in the Materials and Methods section.  Figures 6 and 7 apparently present the same data but without smoothing.

Lines 261-262:  12% moisture content (rather than “humidity”)

Lines 298-299:  Please revise the first sentence for clarity.  “On the longitudinal section of the tested birch wood, the average density decreased going from the tree butt to 20% of the log height.”

Author Response

At the beginning we would like to thank for revision our article. We really appreciate you gave us so many comments. It is extremely important to us to make this paper better and have opportunity to publication in journal such as Forests. We hope that we gave you sufficient answers (below).

Please, note that we add one figure so numeration of the figured changed.

We wrote down the answers with italic.

This article investigates the variation in density of birch wood with position in the log. The results are presented clearly for the most part, are supported by statistical analysis, and should provide useful information for utilization of birch wood. The article is recommended for publication after some improvements in clarity.

Line 38:  Please clarify the meaning of “wood overridden…”

There are several distinctions of the wood zone in the literature, including juvenile, transitional and mature wood (Krahmer 1985, Zobel and Spraque 1998), relating to the variability of wood properties in a radial (Zobel and McElwee 1958, Cown 1980), and vertical distribution (Burdon et al. 2004a). Due to the phases of wood creation, wood zones were separated depending on genetic, environmental and economic conditions: core, transitional and peripheral (Burdon et al.  2004b, Tomczak et al. 2009). In another categorization of wood zones, it was distinguished on the cross-section of the trunk juvenile, mature and overridden wood (found in old trunks) (HeliÅ„ska-Raczkowska 1999). Due to the mechanical properties, mature wood is the most desirable (Jakubowski 2004). Juvenile wood exhibits 50% to 70% strength of mature wood (Kretschmann 1998). In the outer parts of very old trunks, overridden wood occurs, it has inferior properties compared to mature wood (HeliÅ„ska-Raczkowska 1999).

Based on:

Krahmer R.L. 1985. Fundamental anatomy of juvenile and mature wood. A technical workshop: Juvenile wood – what does it mean to forest management and forest products? Forest Products Research Society, 12–29

Zobel B.J., McElwee R.L. 1958. Natural variation in wood specific gravity of Loblolly pine, and an analysis of contributing factors. Tappi Journal 41:158–161.

Cown D.J. 1980. Radiata pine: wood age and wood property concepts. New Zealand Journal of Forestry Science 10:504–507.

Zobel B.J., Spraque J.R. 1998. Juvenile wood in forest trees. Springer Verlag, New York–Berlin.

Burdon R.D., Kibblewhite R.P., Walker J.C.F., Megraw R.A., EVans R., Cown D.J. 2004a. Juvenile versus mature wood: a new concept, orthogonal to corewood versus outerwood, with special reference to Pinus radiata and Pinus taeda. For. Sci. 50(4):399–415.

Burdon R.D., Walker J., Megraw B., EVans R., Cown D. 2004b. Juvenile wood (sensu novo) in pine: Conflicts and possible opportunities for growing, processing and utilization. New Zealand Journal of Forestry 49(3):24–31.

Tomczak A., Pazdrowski W., Jelonek T., GrzywiÅ„ski W. 2009. Jakość drewna sosny zwyczajnej (Pinus sylvestris L.) Część I. Charakterystyka wybranych cech i wÅ‚aÅ›ciwoÅ›ci drewna wpÅ‚ywajÄ…cych na jego jakość. Sylwan 6:363–372

Jakubowski M. 2004. UdziaÅ‚ bielu, twardzieli drewna mÅ‚odocianego i dojrzaÅ‚ego w strzaÅ‚ach sosen zwyczajnych (Pinus sylvestris L.) wyrosÅ‚ych w różnych warunkach siedliskowych. Sylwan 8:16–24

Kretschmann D. E. 1998. Properties of Juvenile wood. Techline. Properties and Use of Wood, Composites, and Fiber Product. United States Department of Agriculture, Forest Service, Forest Products Laboratory. VI-7 Issued 09/98

Helińska-Raczkowska L. 1999. Leksykon nauki o drewnie. Wydawnictwo AR w Poznaniu

Lines 71-72: This mentions determination of the size and range of knots; however, information about knot size is missing from the results section.

The effect of knots on the density in the material tested was determined by measuring the diameter and length of the knots as well as their density from the worse side of the lumber.

Line 86:  “planing” rather than “planning”

We changed it. Thank you for that comment.

Lines 96-97:  Please clarify how defects were taken into account. Were defects included or excluded?

The occurrence of defects mainly in the form of knots had a statistically significant effect on the average density of the large-dimensioned birch wood tested. The average density of wood with defects was higher by 5 kg/m3 compared to the density of wood without defects. The slight difference is due to the significant share of large-sized wood in the knotless zone.

The wood defects such as cracks or knots were included in analysis of wood density distribution, what was described. In results description, we gave information regarding to covering defects with deviations in density on maps presenting density distribution.

Lines 141-142:  The juvenile wood zone is given as a percentage of the log diameter rather than a percentage of cross-sectional area.  This is confusing because line 139 mentions literature values and quotes the area of the cross-section.

The information about juvenile wood in the area of the cross-section was given to order to emphasize a high proportion of that part in all tree trunk volume.

Figures 2,3,4:  The x-axis is lacking units (cm).

Thank you, that is our oversight. It is already corrected.

Figure 5:  Is the curve in this figure based on a mathematical smoothing function?  If so, please provide details in the Materials and Methods section.  Figures 6 and 7 apparently present the same data but without smoothing.

The average densities measured for all 5 logs were used for the relationships shown in Figures 5 and 6. For fig. 6 and 7 the vertical version was also checked, but after the discussion, the presentation of these relationships in the horizontal version seems clearer. In Fig. 5 it is a smoothing line and should be treated as an illustration of the density distribution at the height of the entire birch log. Of course, this is possible in one figure, but would require additional explanations related to the differences between the two interpretations. The height of the crown base was determined on the basis of correlation coefficients table 9 (Between the base of the crown and the top of the tree) and table 10 (Below 0.2 log height and Above 0.2 log height).

Lines 261-262:  12% moisture content (rather than “humidity”)

Thank you. It is already corrected. That was mistake after translating.

Lines 298-299:  Please revise the first sentence for clarity.  “On the longitudinal section of the tested birch wood, the average density decreased going from the tree butt to 20% of the log height.”

Thank you. We corrected that part as it is suggested.

Reviewer 2 Report

In this study, the authors analysed the radial and vertical variations of wood density in the stem of 70 years old trees of Betula pendula from Poland. Juvenile/mature wood transition was assessed through the analysis of the radial variations of fibre length.

The presented results are of great interest, even though only five trees were analysed, but the text is difficult to understand due to language errors.

My main criticism concerns the bibliography, that is seriously deficient. The large majority of references cited are from Polish literature omitting the numerous studies about Betula pendula from Northern countries. Few of the cited studies are post-2000. A simple internet research on "wood density Betula pendula" provides several results that should be cited here.

Before accepting this paper for publication, I consider that most of the text should be rewritten to be more comprehensible. The introduction section should refer to the international literature, and in the discussion, the results should be compared to those of recently published studies.

Among the missing references, the following could be considered:

About the between tree variability of density in birch wood:
- Giagli et al 2019 Stand factors affecting the wood density of naturally regenerated young silver birch growing at the lower altitude of the Cezch Republic region.
- Viherä-Aarnio and Velling 2017 Growth, wood density and bark thickness of silver birch originating from the Baltic countries and Finland in two Finnish provenance trials.
- Liepins, K. and Rieksts-Riekstiòð 2013 Stemwood density of juvenile silver birch trees (Betula pendula Roth) from plantations on former farmlands.

About vertical variations:
- Repola 2006 Models for vertical wood density of scots pine, Norway spruce and birch stems, and their application to determine average wood density

About radial variations:
- Heräjärvi 2004 Variation of basic density and Brinell hardness within mature Finnish betula pendula and B. Pubescens stems

About fiber length:
- Bhat and Kärkkäinen 1981 Variation in structure and selected properties of Finnish birch wood. IV. Fibre and vessel length in braches, sems and roots.

About maps of wood density:
- Longuetaud et al 2016 Within-stem maps of wood density and water content for characterization of species: a case study on three hardwood and two softwood species

About juvenile/mature transition:
- Bonham & Barnett 2001 Fibre length and microfibril angle in silver birch (Betula pendula Roth).

Except the later, all these publications are freely downloadable.

Specific comments:

lines 32-33: Usually, there is no relation between ring width and density for diffuse porous woods. But several authors mention a correlation for other hardwood species like oak. For birch, you could refer to Giagli et al 2019.

lines 34-37: Please rewrite both sentences.

lines 66-67: I don't agree with this statement.

line 85: What is the "core timber"?

lines 93-107: This description is not clear. A figure would help.

line 119: You could have compared this result to Bhat and Kärkkäinen 1981 who reported a mean fibre length of 1.24 mm for Betula pendula.

lines 126-130: This analysis is very interesting. Did you obtain the same age for every tree? Or did you mix the samples from the 5 trees? The result should be compared to the literature, for instance Bonham & Barnett 2001 reported 15 years of juvenile wood.

lines 140-142: You said that the diameter at breast height was 42 cm. According to my calculations, the surface of a 16 cm diameter circle only covers 15% of a 42 cm diameter circle.

lines 144-145: The average density should be compared to the values reported in the literature. The difficulty is that most authors did not provide density at 12% MC but basic density.

lines 175-188: The first sentence is not syntactically correct and the whole analysis remains unclear to me.

line 195: How did you obtain "the distribution of average densities" for the entire birch log?

Figures 2, 3, 4: These figures would need some more detail. Is it the same log that is presented on each figure? What means the x-axis? Is it a distance from the pith in cm? Why did you not use two linear regressions rather than a polynomial one in juvenile wood?

Figures 5 and 6: Ibid. Is it the same log? Why did you invert the x and y axis in both figures? What represents the red line in Fig. 5? Is it a smoothing line? Could you consider represent the smoothed line and the models on the same figure? How did you select or compute the breakpoint between both regressions? Is it somewhat related to crown base height?

Lines 256-283: These maps of density variations are nice, but once again, it should be compared to similar representations in the literature, e.g. Longuetaud et al 2016.

Author Response

At the beginning we would like to thank for revision of our article. We really appreciate you gave us so many comments. It is extremely important to us to make this paper better and have opportunity to publication in journal such as Forests. We hope that we gave you sufficient answers (below).

Please, note that we add one figure so numeration of the figured changed.

In this study, the authors analysed the radial and vertical variations of wood density in the stem of 70 years old trees of Betula pendula from Poland. Juvenile/mature wood transition was assessed through the analysis of the radial variations of fibre length.

The presented results are of great interest, even though only five trees were analysed, but the text is difficult to understand due to language errors.

My main criticism concerns the bibliography, that is seriously deficient. The large majority of references cited are from Polish literature omitting the numerous studies about Betula pendula from Northern countries. Few of the cited studies are post-2000. A simple internet research on "wood density Betula pendula" provides several results that should be cited here.

Before accepting this paper for publication, I consider that most of the text should be rewritten to be more comprehensible. The introduction section should refer to the international literature, and in the discussion, the results should be compared to those of recently published studies.

Among the missing references, the following could be considered:

About the between tree variability of density in birch wood:

- Giagli et al 2019 Stand factors affecting the wood density of naturally regenerated young silver birch growing at the lower altitude of the Czech Republic region.

- Viherä-Aarnio and Velling 2017 Growth, wood density and bark thickness of silver birch originating from the Baltic countries and Finland in two Finnish provenance trials.

- Liepins, K. and Rieksts-Riekstiòð 2013 Stemwood density of juvenile silver birch trees (Betula pendula Roth) from plantations on former farmlands.

About vertical variations:

- Repola 2006 Models for vertical wood density of scots pine, Norway spruce and birch stems, and their application to determine average wood density

About radial variations:

- Heräjärvi 2004 Variation of basic density and Brinell hardness within mature Finnish betula pendula and B. Pubescens stems

About fiber length:

- Bhat and Kärkkäinen 1981 Variation in structure and selected properties of Finnish birch wood. IV. Fibre and vessel length in braches, sems and roots.

About maps of wood density:

- Longuetaud et al 2016 Within-stem maps of wood density and water content for characterization of species: a case study on three hardwood and two softwood species

About juvenile/mature transition:

- Bonham & Barnett 2001 Fibre length and microfibril angle in silver birch (Betula pendula Roth).

Except the later, all these publications are freely downloadable.

Specific comments:

lines 32-33: Usually, there is no relation between ring width and density for diffuse porous woods. But several authors mention a correlation for other hardwood species like oak. For birch, you could refer to Giagli et al 2019.

lines 34-37: Please rewrite both sentences.

lines 66-67: I don't agree with this statement.

We made changes as it was suggested above. We rewrote sentences from lines 34-37 with hope that now they are more understandable. After revising article from the list you prepared, we couldn’t also agree with statement from lines 66-67. Therefore, changes were necessary.

line 85: What is the "core timber"?

This is 70 mm thick sawn timber containing a pith.

lines 93-107: This description is not clear. A figure would help.

Because we want to make article clear for everyone, and used methods are not popular, we prepared the diagram - fig. 1.

line 119: You could have compared this result to Bhat and Kärkkäinen 1981 who reported a mean fibre length of 1.24 mm for Betula pendula.

Thank you for that comment, but we decided to complement that part with more contemporary literature.

lines 126-130: This analysis is very interesting. Did you obtain the same age for every tree? Or did you mix the samples from the 5 trees?

For testing the width of annual increments and the length of wood fibers, 3 logs were selected, from which cut strips at the breast height along the northern radius. The presented and discussed results are average and represent the properties of the tested material.

The result should be compared to the literature, for instance Bonham & Barnett 2001 reported 15 years of juvenile wood.

We made comparison as you suggested.

lines 140-142: You said that the diameter at breast height was 42 cm. According to my calculations, the surface of a 16 cm diameter circle only covers 15% of a 42 cm diameter circle.

It was our mistake, we made calculation again and corrected description.

lines 144-145: The average density should be compared to the values reported in the literature. The difficulty is that most authors did not provide density at 12% MC but basic density.

The basic density determination requires wood to be at moisture content 0%. For large elements with a length of 1.5 m and a thickness of 70 mm, it seems impossible to maintain such humidity during testing. Density determination with a non-destructive method was made in lab with constant conditions. It let to keep moisture content of wood at the same level during testing.

lines 175-188: The first sentence is not syntactically correct and the whole analysis remains unclear to me.

Tukey test of the factor "distance from the core" for the distribution of average densities over the cross section was made for the entire birch log. Comment “For comparison, the distribution of average densities across the cross section at the height of the breast height and ¼ of the height of the birch log was examined, which is shown in Figure 3 and 4. At the dbh and at ¼ of the height of the birch logs in cross-section, the average density of wood increased along the radius of the tree from the core to a distance of about 8 cm from the core, and then there was very little variation in density” regards to density at breast high and density at the ¼ of the log height.

line 195: How did you obtain "the distribution of average densities" for the entire birch log?

The average density for each log was obtained by adding at designated points in the longitudinal (every 150 mm) and transverse (every 20 mm) directions all measured densities and then dividing the sum by the number of measurements made. In the further analysis of the obtained variations (13-14 bales from each log, 69 bales and 8320 density measurements in total), statistical measures were used, assuming that the results obtained were normal (Kolmogorov-Smirnov test).

Figures 2, 3, 4: These figures would need some more detail. Is it the same log that is presented on each figure? What means the x-axis? Is it a distance from the pisth in cm? Why did you not use two linear regressions rather than a polynomial one in juvenile wood?

On the x axis the diameter of tested elements, divided into two parts measured symmetrically from the log pith (radius). The presented density distributions refer to average densities measured for the cross-section, at the height of the breast and ¼ of the height of the log, as the average values for all 5 tested logs. The use of the 2nd degree polynomial function to determine the density distribution of juvenile wood resulted from obtaining a high coefficient of determination, much higher, compared to linear functions.

Figures 5 and 6: Ibid. Is it the same log? Why did you invert the x and y axis in both figures? What represents the red line in Fig. 5? Is it a smoothing line? Could you consider represent the smoothed line and the models on the same figure? How did you select or compute the breakpoint between both regressions? Is it somewhat related to crown base height?

The average densities measured for all 5 logs were used for the relationships shown in Figures 5 and 6. For fig. 6 and 7 the vertical version was also checked, but after the discussion, the presentation of these relationships in the horizontal version seems clearer. In Fig. 5 it is a smoothing line and should be treated as an illustration of the density distribution at the height of the entire birch log. Of course, this is possible in one figure, but would require additional explanations related to the differences between the two interpretations. The height of the crown base was determined on the basis of correlation coefficients table 9 (Between the base of the crown and the top of the tree) and table 10 (Below 0.2 log height and Above 0.2 log height).

Lines 256-283: These maps of density variations are nice, but once again, it should be compared to similar representations in the literature, e.g. Longuetaud et al 2016.

We looked at this article quite thoroughly. It does not apply to birch wood and analyzes density changes for other wood species, as well as other relationships related to its moisture content.

Reviewer 3 Report

It was important to accumulate the fundamental density data for European birch. However, since there is a lot of data, it is better to summary and discuss more. thus, there are few comments.

・Figure 1:It was a critical problem to choose the juvenile zone and mature zone. You selected the line about tree age 25 to calculate the regression carve. Is there a micrograph near 25 tree age? Or, what is the regression results when evaluated using other annual ring data?

・Line 159,181: Does calculating an approximate curve make sense? In particular, what does the x-square equation mean? What is the definition of core (measuring line) in Figure 2?

・You select the data from breast height and 1/4 of the log height. What made you choose these height?

・Figure 5: What is the correspondence between average density and tree appearance ?

・Figure 8:Overwrite the juvenile and mature wood line in Figure 8, and discuss the correlation between the wood density and the maturity of cell (wood anatomy).

・In conclusion section: This section has no discussion. It was merely summarized in your work. Is it possible to proceed the discussion with the your obtained data?

Author Response

At the beginning we would like to thank for revision our article. We really appreciate you gave us so many comments. It is extremely important to us to make this paper better and have opportunity to publication in journal such as Forests. We hope that we gave you sufficient answers (below).

Please, note that we add one figure so numeration of the figured changed.

It was important to accumulate the fundamental density data for European birch. However, since there is a lot of data, it is better to summary and discuss more. thus, there are few comments.

 ãƒ»Figure 1:It was a critical problem to choose the juvenile zone and mature zone. You selected the line about tree age 25 to calculate the regression carve. Is there a micrograph near 25 tree age? Or, what is the regression results when evaluated using other annual ring data?

We did not prepare any micrographs.

Assuming that the boundary between juvenile wood and mature wood is further from the pith, the equation describing changes in fibers length has a smaller fit (lower correlation coefficient).

・Line 159,181: Does calculating an approximate curve make sense? In particular, what does the x-square equation mean? What is the definition of core (measuring line) in Figure 2?

The core was used in meaning the pith. We changed that term to make description more understable if you found it confusing.

X square has no specific meaning. It is part of equation.

・You select the data from breast height and 1/4 of the log height. What made you choose these height?

That high was chosen based on literature data.  We based on two articles:

Hakkila, P. Investigations on the basic density of Finnish pine, spruce and birch wood. Communicationes Instituti Forestalis Fenniae 1966, 61 (5), 1 – 98.

Hakkila, P. Wood density survey and dry weight tables for pine, spruce and birch stems in Finland. Communicationes Instituti Forestalis Fenniae 1979, 96 (3), 1 – 59.

・Figure 5: What is the correspondence between average density and tree appearance ?

Such analyzes were not carried out directly. The selection of research material - in addition to the selection of a strictly defined habitat type - fresh forest, harvested logs were characterized by a straight arrow and a symmetrical crown. After sawing, taking into account geographical directions, a selection of logs was made, which after the drying process was subjected to a qualitative assessment. It was based on Polish standard PN-72/D-96002 titled: "General purpose deciduous timber", due to the lack of a European standard specifying quality classes based on appearance sawn timber and standards for the visual strength classification of hardwood construction timber. In the tested material, the proportion of class III logs was 70%, class II was 27%, and the so-called waste 3%. There was no class I in the tested material.

・Figure 8:Overwrite the juvenile and mature wood line in Figure 8, and discuss the correlation between the wood density and the maturity of cell (wood anatomy).

The correlation between the wood density and the maturity of cell was described in article next to fig. 8.

The proposal of separating the juvenile wood zone along the entire length of the log and determining the structure of its tissue and neighboring structures is a valuable inspiration for conducting further research in this field, which the authors of the article thank very much.

・In conclusion section: This section has no discussion. It was merely summarized in your work. Is it possible to proceed the discussion with the your obtained data?

Discussion of the obtained results was made with reference to the literature during the discussion of relationships depicting the density of birch wood with separation of mature and juvenile wood zones. This continuous, non-destructive density measurement on the longitudinal and transverse cross-section, resulting in a map of density distribution in birch wood logs, has not been described in the literature so far.

The studies cited in the literature review concern various aspects related to point in a limited area, and are associated with destructive research methods. In the presented study, for the purpose of density tests, on the longitudinal and transverse cross-section of birch wood logs and the statistical analysis of its variability, the quality parameters used. This initial verification of quality parameters used for structural softwood was used to assess diminutive deciduous wood. A non-destructive testing method was used to conduct the density distribution analysis. The observations contained in this study may become useful for the development of further studies on various other species of deciduous trees, e.g. plantation trees, which in the future could expand the raw material base of construction timber.

Round 2

Reviewer 3 Report

I am satisfied with the change of the author.

Good works and I hopes for research approaching essence considering wood anatomy and wood physics in next research.